# Diagnosis and Treatment of Post-Prostatectomy Lymphedema: What's New?

Lorenzo Maria Giuseppe Bianchi [1] , Giovanni Irmici [1] , Maurizio Cè [1], Elisa D'Ascoli [1] , Gianmarco Della Pepa [1] , Filippo Di Vita [2], Omar Casati [2], Massimo Soresina [3], Andrea Menozzi [3], Natallia Khenkina [1] and Michaela Cellina [4,*]

1   Postgraduation School in Radiodiagnostics, University of Milan, 20122 Milan, Italy
2   Postgraduation School in Plastic Surgery, University of Milan, Via Festa del Perdono, 7, 20122 Milan, Italy
3   Plastic Surgery Department, Fatebenefratelli Hospital, ASST Fatebenefratelli Sacco, 20121 Milan, Italy
4   Radiology Department, Fatebenefratelli Hospital, ASST Fatebenefratelli Sacco, 20121 Milan, Italy
*   Correspondence: michaela.cellina@asst-fbf-sacco.it

**Abstract:** Lymphedema is a chronic progressive disorder that significantly compromises patients' quality of life. In Western countries, it often results from cancer treatment, as in the case of post-radical prostatectomy lymphedema, where it can affect up to 20% of patients, with a significant disease burden. Traditionally, diagnosis, assessment of severity, and management of disease have relied on clinical assessment. In this landscape, physical and conservative treatments, including bandages and lymphatic drainage have shown limited results. Recent advances in imaging technology are revolutionizing the approach to this disorder: magnetic resonance imaging has shown satisfactory results in differential diagnosis, quantitative classification of severity, and most appropriate treatment planning. Further innovations in microsurgical techniques, based on the use of indocyanine green to map lymphatic vessels during surgery, have improved the efficacy of secondary LE treatment and led to the development of new surgical approaches. Physiologic surgical interventions, including lymphovenous anastomosis (LVA) and vascularized lymph node transplant (VLNT), are going to face widespread diffusion. A combined approach to microsurgical treatment provides the best results: LVA is effective in promoting lymphatic drainage, bridging VLNT delayed lymphangiogenic and immunological effects in the lymphatic impairment site. Simultaneous VLNT and LVA are safe and effective for patients with both early and advanced stages of post-prostatectomy LE. A new perspective is now represented by the combination of microsurgical treatments with the positioning of nano fibrillar collagen scaffolds (BioBridgeTM) to favor restoring the lymphatic function, allowing for improved and sustained volume reduction. In this narrative review, we proposed an overview of new strategies for diagnosing and treating post-prostatectomy lymphedema to get the most appropriate and successful patient treatment with an overview of the main artificial intelligence applications in the prevention, diagnosis, and management of lymphedema.

**Keywords:** lymphedema; prostatectomy; lymphedema new treatments; robotic surgery; deep-learning segmentation systems

## 1. Introduction

Lymphedema refers to tissue swelling caused by excessive retention of lymphatic fluid in the interstitial compartment as the result of an imbalance between the generation of lymph and its drainage into the systemic circulation [1].

This condition can affect different areas of the body such as arms, legs, genitals, face, neck, chest wall, and oral cavity [1].

Recent evidence has shown that lymphedema is a multifactorial process including lymphatic stagnation and chronic inflammation, associated with abnormal adipose tissue growth and fibrosis. Adipocytes can easily absorb free fatty acids, one of the numerous

metabolites present in the lymph, leading to the development of hypertrophic adipose tissue [2]. Hypertrophic fat lobules constrict and compress lymphatic capillaries by interfering with the transport of fluids and lipids, leading to a vicious cycle in which increased peripheral fat deposition further impedes lymphatic drainage [3]. Increased release of pro-inflammatory cytokines (TNF-$\alpha$, IL-6, MCP-1, and IL-8), that enhance immune cell recruitment and their pro-inflammatory polarization, is a result of this adipose tissue dysregulation, metabolic stress, and dysfunction of lymphatic vessels [4]. Fibrotic tissue degeneration and skin thickening caused by hyperkeratosis are the final pathologic features of late-stage lymphedema, accompanied by an increased risk of recurrent local infections, and development of cellulitis, ulcers, fissures, and, very rarely, cutaneous angiosarcoma [1].

Lymphedema can be primary or secondary, depending on the etiology [1]. Primary lymphedema is a rare condition caused by genetic abnormalities that impair lymph channel development, with an incidence about double in females compared to males [5], and is categorized into three groups according to the age of the onset: congenital lymphedema or Milroy disease (<2 years), familial lymphedema praecox (around puberty), and lymphedema tarda (>35 years) [6,7].

The most frequent cause of secondary lymphedema is heartworm disease, which affects more than 90 million individuals worldwide, mostly in low-income regions, and is caused by infection with the parasite Wuchereria bancrofti. Mature heartworms lodge in the lymphatic ducts interfering with lymphatic circulation [8]. The leading cause of secondary lymphedema in Western nations is cancer or cancer-related treatments, such as radiation therapy, lymph node metastases, and regional lymph node surgery; it usually occurs 12–18 months after damage to the lymphatic system [9]. The incidence of secondary lymphedema is difficult to quantify, as it depends on the site of the tumor and the type of treatment, as well as the individual predisposition. Cormier et al. [10] in their systematic review reported the incidence of cancer-related lymphedema showing a large gap between the different types of cancer and a wide range within each type: breast cancer (6–63%), melanoma (1–66%), gynecological malignancies (0–73%), and genitourinary malignancies (1–23%), with an overall incidence of 15.5%. According to this review, patients undergoing lymph node pelvic dissections (22%) and local radiation therapy (31%) had an increased risk of lymphedema.

A total of 1,414,259 new cases of prostate cancer and 375,304 related deaths were reported in 2020 globally [11]. Advances in therapy have led to an increase in survival, which unfortunately has not always led to a parallel improvement in the quality of life. One of the most frequent complications of surgical and/or radiotherapy treatment of prostate cancer is lymphedema of the lower limbs, groin, and genital area, with joint stiffness, hyperkeratosis, skin dyschromia, altered sensitivity, and heaviness of the limbs, associated with an increased risk of infection [12]. Despite its high incidence, these complications have not received the same attention as lymphedema in breast cancer. One of the main reasons may be that while patients with breast cancer place more emphasis on the functional and aesthetic elements of lymphedema, in patients with prostate cancer, the main concern is sexual and urogenital functions [13]. In a recent systematic review, Clinckaert et al. [14] showed that in prostate cancer patients, the rate of secondary lymphedema of the lower limbs ranged from 0% to 14% in subjects treated with pelvic lymph node dissection and from 0% to 8% in patients treated with pelvic lymph node radiotherapy. Furthermore, the prevalence was higher (between 18 and 29%) in the subgroups who had irradiated pelvic lymph nodes after lymph node dissection, indicating that the combination of surgery and irradiation results in significantly higher rates of lymphedema. Deriving the incidence rates of penile and scrotal lymphedema is more difficult because, in most studies, it is typically considered together with lower extremity lymphedema; it was calculated from 0.5 to 1.5% in patients who underwent surgery [15,16], while in the case of radiotherapy, the prevalence was between 0 and 6% with the highest prevalence of 22% in case of radiotherapy following node dissection, the same result obtained in the case of the lower limbs lymphedema [14].

While in the past, the diagnosis and therapy were based on a clinical approach and multiple non-invasive tests such as tape measurement [17], bioelectrical impedance analysis [18], or volumetry [19], today, imaging has assumed an essential role in the assessment of severity, differential diagnosis, and therapeutic planning. Lymphoscintigraphy has been the standard reference for many years; it is a method that consists in injecting a radiolabeled colloid into the distal portion of an edematous limb, between the first and second finger, and then photographing the progression of the tracer through the lymphatic vascular system with a gamma camera. This technique offers insight into the anatomy and function of the lymphatic system [20]. Recently, new methods have been developed for the study of lymphedema, in particular magnetic resonance lymphography (MRL), which can be performed without the use of contrast agents (NCMRL) or through injection of gadolinium into the web spaces between the fingers and toes (contrast-enhanced magnetic resonance lymphography (CEMRL)) [21].

Traditionally, patients with lymphedema were treated with conservative therapies such as bandaging, compression, and manual drainage. Recently there have been substantial innovations in both medical therapy and surgical procedures, with a significant improvement in the patient's quality of life. For example, robots can be used for drainage, compression treatment, and microsurgical approaches; new experimental drugs have been introduced as Captopril and Tacrolimus.

This article aims to describe the new diagnostic and therapeutic frontiers of lymphedema in patients treated for prostate cancer.

## 2. Imaging

Technological advances in biomedical imaging have opened new possibilities in the diagnosis and treatment of LE. In this review, we will focus on these lymphatic imaging techniques. Different techniques are available to visualize the structure and functions of peripheral and central lymphatic vessels. These techniques can evaluate the lymphatic vessels either directly, showing abnormal lymphatic vessel dysfunction, or indirectly, showing sequelae of lymphatic vessel dysfunction, such as the development of fibrosis or other pathologic tissue features [22].

Historically, lymphography (or lymphangiography) has been the method of choice for imaging the lymphatic system: it was performed under fluoroscopy by injecting contrast media directly into lymph nodes or lymphatic vessels in the subcutaneous tissue or intramuscularly [23].

Later, for years, lymphoscintigraphy has been considered the reference technique for diagnosis; a small quantity of radioactive protein dye is injected into the region into the interdigital spaces of the affected limb, and a gamma camera is used to take images of that limb to track the dye's passage through the lymphatic system. This procedure is still regarded as the gold standard for evaluating the lymphatic system [24].

Other techniques can have a role in edematous limb management; for example, ultrasonography (US) and computed tomography can be used to demonstrate a venous insufficiency etiology or rule out deep vein thrombosis [25]. Computed tomography can help in assessing a quantitative skin and subcutaneous tissue volume involvement [26] or finding a subsequent cause of lymphedema.

In recent years, new imaging modalities have been proposed to help lymphedema with evaluation.

### 2.1. MRL

MRL is a relatively new technique that combines morphological and functional data in a single examination and can play an important role in planning the best therapy for the patient [27]. It can be acquired with (CEMRL) [28,29] and without (NCMRL) [30,31] contrast medium administration into the interdigital spaces. NCMRL is a non-invasive imaging technique that plays a main role in the severity assessment, treatment planning, and follow-up of secondary lymphedema, as it provides information regarding different

key aspects, such as the symmetry/asymmetry of lower or upper limbs involvement, the accurate measurement of the size of the affected/unaffected limbs, as well as the thickness and composition of the subcutaneous fat and lymphedema severity [32].

The execution of NCMRL requires a high-field MRI scanner, generally, a 1.5-Tesla, with a multielement body coil (usually a combination of a multi-channel phased array body coil for the lower abdomen to study the iliac and inguinal lymphatic plus a lower limb dedicated coil) [33]; patients are asked to suspend the lymphatic drainage for 48 hours and use elastic stockings or bandages for 24 h [21].

NCMRL protocols are still evolving, but the base of this examination consists of heavily T2-weighted sequences, with very long both Repetition Time (TR) and Echo Time (TE), usually performing 3D sequences (to create rotating 360° 3D post-processed images and maximum intensity projection (MIP) reconstructions), with very long TR/TE ratio to assess the extent and distribution of the lymphedema [34] (Figure 1).

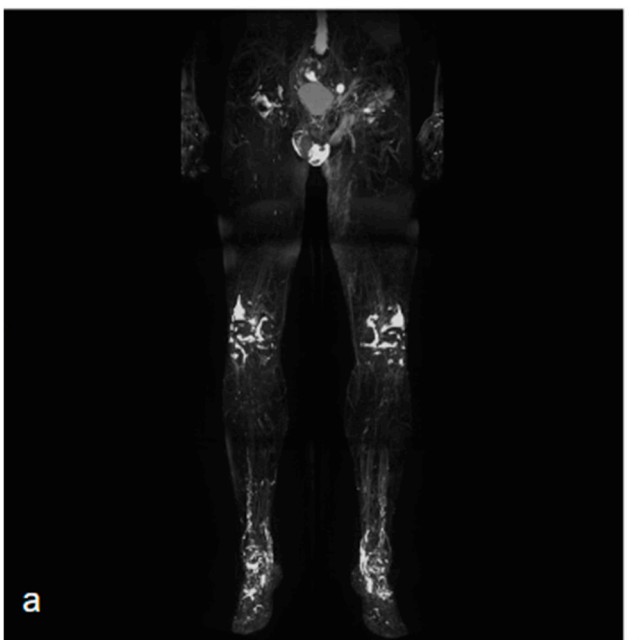 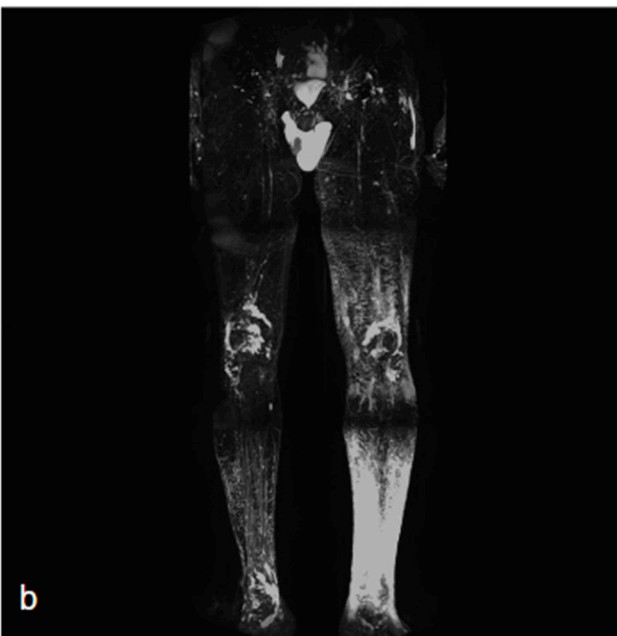

**Figure 1.** Two different NCMRL. On the left (**a**), a patient with mild post-prostatectomy lymphedema with a left iliac lymphocele associated. On the right (**b**), mild–severe lymphedema is more represented in the left limb, with scrotal localization.

CEMRL is performed with gadolinium administration in the interdigital web spaces with fat sat 3D fast spoiled gradient-echo T1 sequences. To suppress the venous signal for a better distinction of lymphatic vessels, some authors suggested the concurrent administration of intravenous gadolinium [35,36].

On coronal T2-weighted images, lymphedema presents as an epifascial distribution with a high fluid-like signal intensity, with changes in the subcutaneous fat tissue that is hypertrophic with honeycombing modifications [9,21].

CEMRL is able to provide functional information on the lymphatic function and lymph nodes contrast uptake, but requires contrast medium administration; NCMRL instead does not need contrast medium, but cannot depict normal or hypoplastic lymphatic structures and does not give any functional data [37].

### 2.2. Near-Infrared Fluorescence Imaging

An effective substitute for assessing lymphatics is near-infrared fluorescence imaging, which uses a contrast media injected intradermally instead of a radionuclide to access the primary lymphatics beneath the epidermis, helping to assess lymphatic anatomy and function [38] (Figure 2).

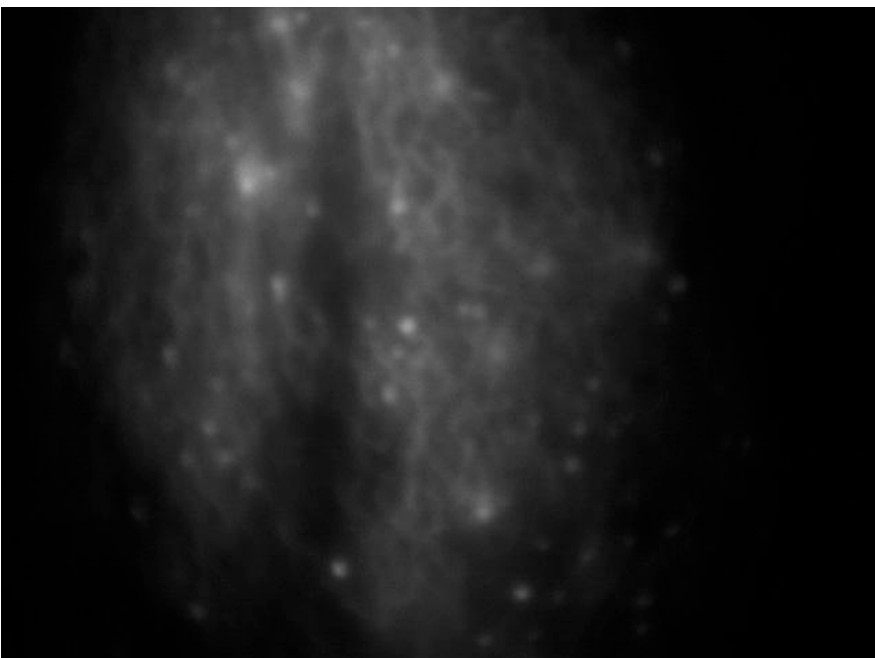

**Figure 2.** Intraoperative ICG lymphography for lower limb lymphatic vessel evaluation in a patient with post-prostatectomy secondary lymphedema.

The main benefit consists of the acquisition of real-time images.

When a substance is exposed to the light of a different wavelength, it emits light with a specific wavelength, which causes the phenomenon known as fluorescence. Irradiated light is called excitation light while emitted light is called fluorescence light [39,40].

Two dyes are usually used in diagnostics: fluorescein sodium, which is excited by visible light, and indocyanine green (ICG), which is excited by near-infrared light. [22].

ICG lymphography is performed by injecting this fluorescent dye into the interdigital web spaces; after 4 h, a photodynamic eye camera equipment can be used to capture fluorescence images of lymphatic vessels to a maximum depth of 2 cm; the fluorescence images are digitalized for real-time display [21,41] and can be used also for sentinel lymph node detection in cancer interventions [42].

ICG emits very little fluorescence, but when coupled with plasma proteins and exposed to the proper wavelength of light, it becomes more visible. A study using fluorescence lymphography found that 25% of patients had four aberrant patterns: diffuse sparkling, dilated lymph channels with proximal obliteration, prolonged dermal backflow, and dermal backflow in the foot. Only in one instance did multiple patterns appear in the same limb [43]. ICG fluorescence is a good qualitative test, according to the authors.

Yamamoto et al. [44–46] developed a severity score of the dermal backflow for upper and lower limbs secondary lymphedema with three different stages.

Two different studies proposed by Mihara et al. [47] and Akita et al. [48] comparing ICG lymphography with lymphoscintigraphy stated that ICG lymphography has a higher accuracy, useful for surgical workup. Moreover, ICG lymphography does not involve radionuclides, is cheaper than lymphoscintigraphy, and seems to be effective in the early identification of lymphedema.

## 3. Conservative Treatments

### 3.1. Mechanical and Physical Therapies

The first choice for conservative treatment of lymphedema is currently complex decongestive therapy (CDT); this type of approach only acts on the symptoms and not on the causes. CDT is divided into two phases; the aim of the first phase is maximum limb volume reduction and is based on skincare, manual lymphatic drainage (MLD), compression

therapy, and exercises [49,50]. MLD promotes the absorption of fluids and proteins from the interstitium into the lymphatic capillaries, increases the contractility of the lymphatic collectors, and increases the amount of fluid returning to the venous system [51]. Compression therapy is performed with multilayer bandages, adjustable compression devices, and elastic garments and favors the reabsorption of interstitial liquids, improves muscle pumping and venous return, and promotes the release of anti-inflammatory mediators. The second phase aims to maintain and optimize the results obtained and is performed with the application of elastic garments, exercises, skincare, and MLD when necessary [50]. Compression garments, which come in a range of sizes, designs, and degrees of elasticity, can be worn for an extended period while keeping the patient completely mobile. Its success, however, is primarily dependent on its fit to the patient's anatomy, and its administration is especially difficult in patients with a limited range of motion [52]. While stationary pumps help with lymph drainage, effective systems that are wearable while conducting daily tasks are still not widespread. In this scenario, new robotic gadgets are the future of compression devices. Graduated pressure can be administered along the leg using microcontroller-controlled pneumatics devices with pressure feedback, such as the one designed by Rosalia et al., which allows graduated pressure to be applied along the limb [53]. The inflatable compression sleeve is placed between a washable, skin-protective internal envelope and a robust external layer. The pump and valve system can be disengaged from the sleeve after the desired pressure is attained, allowing the user to continue moving. The latest innovation in the development of soft robotic sleeves is clearly reflected in the project of Gao et al., who exploited the potential use of air microfluidic chips [54]. They developed a prototype of a soft robotic sleeve with a built-in valve-free device that provides a delayed, gradient pneumatic actuation that mimics the massage movements of the MLD, using microfluidic fundamentals and in-house manufacturing processes. Thanks to its cutting-edge technological equipment, this device has the advantage of being simple and lightweight for ease of use. Mukherjee et al. [55] observed that lymphatic contractile activity may be appropriately regulated utilizing a spatiotemporally variable oscillatory pressure wave applied in vivo to active collecting lymphatics during intermittent pneumatic compression therapy.

Electrical stimulation could represent a new therapeutic approach; more studies are needed, but electrical stimulation has been shown to influence the pathophysiological mechanisms at the base of lymphedema development, related to the sensitivity of the cells to modifications in electric fields, as the function of membrane proteins is influenced by ionic gradients through cell membranes creating a trans-membrane voltage [56]. An in vitro study by Kajiya et al. showed that electrical stimulation results in the activation of proteins such as ERK and p38, which are associated with the migration and proliferation of lymphatic endothelial cells) and could thus help restore lymphangiogenesis and promote lymphatic drainage [57]. Other studies have shown that electrical stimulation results in the release of trophins such as VEGF, leading to neo-angiogenesis and thus may favor ulcer repair [58] and FGF-1 and FGF-2, which stimulate the transdifferentiation of fibroblasts into myofibroblasts, promoting the reversibility of the fibrosis that characterizes the advanced stages of lymphedema [59].

### 3.2. Pharmacological Therapies

Pharmacological approaches have also been tested in recent years. Brown et al. have recently demonstrated that local inhibition of the renin–angiotensin system (with a topical angiotensin-converting enzyme inhibitor (5% captopril)) is beneficial in animal models of lymphedema [60]. This could be explained by the fact that dermal fibrosis and fibrotic occlusion of lymphatics are histopathological features of lymphedema. The renin–angiotensin system has a key role in kidney and cardiovascular mechanics and also in fibrosis regulation in different organs and systems through the modulation of the intracellular TGF-β1 signaling [61]. A further encouraging pharmacological treatment was investigated by Hansen et al. who evaluated the use of topical tacrolimus in women with stage I or II Breast cancer-related lymphedema (BCRL) with good results in relieving BCRL in terms

of improved arm volume, L-Dex, and HRQoL [62]. Mehrara et al. evaluated the efficacy of a monoclonal IL4/IL13 neutralizing antibody (QBX258) in women with BCRL as recent studies have suggested that Th2 cells play a key role in the pathology of secondary lymphedema by processing cytokines such as IL4 and IL13; they demonstrated that the treatment improved skin stiffness and quality of life. It also reduced epidermal thickness, the number of proliferating keratinocytes, type III collagen deposition, mast cell infiltration, and the expression of Th2-inducing cytokines in edematous skin [63]. Another trial by Rockson et al. supports the benefit of targeted anti-inflammatory therapy, demonstrating that ketoprofen treatment acts on microlymphatic dysfunction within the skin, resulting in a significant cutaneous thickness decrease ($62.1 \pm 8.4$ mm pre-therapy versus $27.4 \pm 5.6$ mm post-therapy, p: 0.0006); the mechanism of this effect is related to its negative action on leukotriene B4 (LTB4) production through inhibition of 5-lipoxygenase [64].

## 4. Surgical Treatment

Surgical management of lymphedema is typically reserved for more severe cases and can be divided into physiological and debulking procedures. The goal of surgical treatment is to improve the functional status of patients (Figure 3).

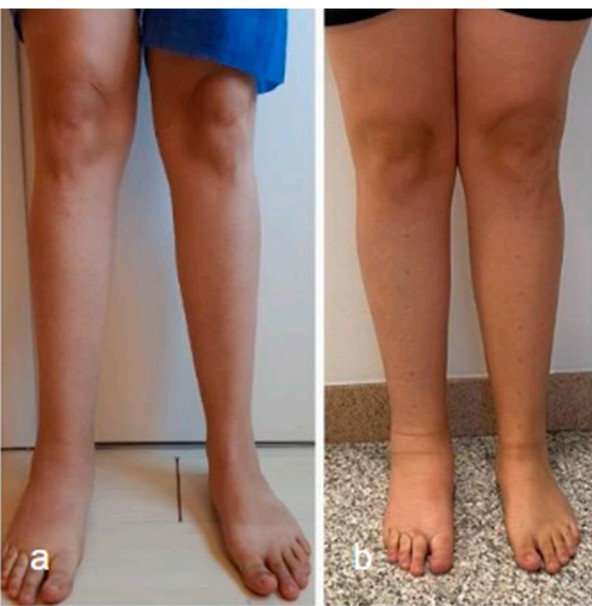

**Figure 3.** Lower limb conditions 9 months post-surgery (**a**) compared to preoperative conditions (**b**) in a patient with secondary lymphedema treated with VLNT in the right knee (using right lateral thoracic flap) and BioBridge™.

Ablative procedures are used to surgically remove tissue excess, reduce morbidity, and restore limb function. These procedures are typically indicated in the more advanced stages of secondary lymphedema or in the case of primary lymphedema where the absence of lymphatic vessels excludes the possibility of physiological treatments [65].

With the progression of microsurgical techniques and knowledge of the pathophysiology of lymphedema, procedures have been developed to increase the number of lymphatic pathways to promote the forward flow of lymph. These procedures, called physiological procedures, are typically used in the early stages of the disease and consist of lymphovenous anastomosis (LVA) and vascularized lymph node transplantation (VLNT) [66]. Recently, the addition of collagen nanofibrillar scaffolds to promote the formation of new lymphatic vessels has improved the outcome of these procedures [67].

### 4.1. Ablative Procedures

Ablative operations attempt to reduce the volume of the affected limb, enhance the functional status, and facilitate decongestive therapy by removing extra skin and sub-

cutaneous tissue. These strategies are applied in patients with late-stage II and stage III lymphedema who are not eligible for other surgical operations because of excessive adipose tissue and fibrosis. These techniques can also be used together with physiologic procedures to address fat tissue and fibrosis [68]. Liposuction is a minimally invasive, low-risk procedure and is unlikely to have further negative effects on lymphatic drainage; nevertheless, it has a positive cosmetic impact on patients. After the treatment, a significant volume reduction in the affected limb is seen almost immediately, but this is not curative, and patients are forced to continue conservative therapy and compression to maintain results [69]. This technique may also be used concurrently with LVA or VLNT to reduce limb size in patients whose lymphatic function has been restored [70,71].

The "Charles procedure" involves extensive adipose tissue removal up to the deep fascia level, combined with reconstruction with skin transplants. This intervention is associated with the disruption of the native distal lymphatic vessel system and can sometimes increase distal limb lymphedema. The high risk of complications, including graft rupture, ulceration, and hypertrophic scarring, together with the poor cosmetic outcome, limit this procedure to very severe cases that are ineligible for other therapies [72].

A debulking operation called phased subcutaneous excision, which preserves the skin, blood vessels, and nerves while removing excess subcutaneous tissue, is another possibility. Compared to the Charles technique, it is less drastic and does not call for skin graft reconstruction. Although this procedure carries high morbidity, it can be useful in patients with advanced disease, with the aim of partially or completely restoring mobility [73].

### 4.2. LVA

LVA is a microsurgical procedure that aims to bypass proximal lymphatic obstruction by diverting lymphatic fluid into the venous system. Patient selection is crucial for successful outcomes: good candidates for the procedure are those presenting with stage I or II lymphedema and minimal improvement after conservative measures.

Preoperative imaging using indocyanine green (ICG) lymphography aids in the identification of functioning lymphatics and the selection of incision placement. Under microscope magnification, the skin is incised, and lymphatic channels are dissected and cut proximally before being anastomosed to a nearby venule of appropriate size using various techniques, involving different anastomoses like end-to-end, end-to-side, or side-to-end. The competence of the surgeon and meticulous tissue control is critical to success due to the small size of lymphatic channels, often 0.5–0.8 mm in diameter. To ensure that the entire thickness is pierced and to prevent back pressure, the vessel can be floated with heparinized saline or stented with a 6-0 Prolene suture. The patency of the anastomosis can be confirmed using ICG lymphography or isosulfan blue injection. Typically, three to four LVA procedures are performed per affected limb, and postoperatively, patients are encouraged to walk but avoid vigorous exercise.

Outcomes of LVA have shown promising results, with significant reductions in postoperative limb volume at 3, 6, and 12 months after LVA and an important reduction in episodes of cellulitis in the affected extremity. However, Technical challenges and a paucity of long-term outcomes in the literature are drawbacks. Patients should be informed that LVA is not a curative treatment and that fibrosis or anastomosis failure can cause the condition to return [74].

### 4.3. VLNT

VLNT is usually recommended in lymphedema stages II and III, for subjects with significant dermal backflow without any functioning lymphatic, affected by cellulitis, and in those who have had no benefits from at least 12 months of complete decongestive therapy (CDT) [66]. In this group of patients, results have shown a 30–60% reduction of excess volume compared to the contralateral unaffected limb; overall quality of life and functional status have been consistently improved, while skin infection rates and need for CDT were decreased. Lymphedema of the upper extremities responded better than the

lower extremities (74.2 vs. 53.2%), but there was no difference in lymph node placement proximal to distally on the extremity (proximal: 76.9% vs. distal: 80.4%) [75].

The procedure consists in harvesting healthy lymph nodes with their vascular supply and transferring them to the affected area using a surgical microscope. It is mandatory to maintain the vascularity of the lymph nodes for better lymphatic function. The newly transplanted lymph nodes act by absorbing lymphatic fluid and directing it into the vascular system. These lymph nodes secrete growth factors (such as the vascular endothelial growth factor (VEGF)) that stimulate the generation of new lymphatic ways and channels.

In VLNT, nodes harvested from the groin, thoracic, submental, and supraclavicular regions are used, but mesenteric lymph node transfer and omental transfer have been reported as well. In 2018, a systematic review included 24 studies involving 271 vascularized lymph node transfers, inguinal lymph nodes were the most used donor site, followed by lateral thoracic lymph nodes [75].

Most of the time, supraclavicular lymph nodes based on the transverse cervical artery and vein and the superficial inguinal lymph nodes based on the superficial circumflex iliac artery are used. For lower extremity lymphedema, the ankle and the groin are commonly considered the best recipient sites; when the groin is concerned, it is important to remove scar tissue caused by previous surgery and radiation before proceeding to VLNT. Superficial circumflex iliac vessels are generally utilized for anastomosis. When lymphedema occurs distal to the groin, the ankle is used as a recipient site [76]. Gravitational forces pull fluid into the ankle, so lymph nodes positioned here facilitate drainage. In these cases, anterior tibialis or dorsalis pedis arteries are often used for anastomosis. The knee is less commonly used as a recipient site; in this case, the medial genicular branches or the saphenous vessel branches are considered for vascular anastomosis [77,78].

In comparison to other lymph node donor locations, lateral thoracic lymph nodes were the least efficient and had the highest rate of complications (27.5% versus 10.3% in inguinal and 5.6 % in supraclavicular harvesting) [75].

A recent study by Leppäpuska et al. has shown the efficacy of a new treatment combining a pro lymphangiogenic growth factor vector inducing VEGF-C with VLNT in the treatment of secondary lymphedema due to breast cancer: the most promising findings were a 46% reduction in excess arm volume after 24 months of follow-up [79].

One rare but serious complication of VLNT is iatrogenic lymphedema, which has been described in a few cases [66,80]. To minimize risks, it is crucial to harvest only expendable lymph nodes and preserve those that drain the extremities. Reverse lymphatic mapping is a technique introduced by Dayan et al. to differentiate between these lymph nodes. Technetium is injected into the distal web spaces of the affected extremity before VLNT, and a gamma probe is used to identify the critical lymph nodes that demonstrate technetium uptake. A near-infrared camera is employed during harvest after ICG injection into the trunk near the donor lymph nodes to assess that only the lymph nodes draining the trunk are included with the donor flap. This technique is useful to minimize risks due to the considerable variations in anatomy and lymphatic drainage patterns [81].

### 4.4. Aligned Nano Fibrillar Collagen Scaffolds (BioBridge™)

None of the procedures mentioned above could restore the non-functional lymph vessels along the entire length of the limb. To address this challenge, some groups have begun incorporating nanofibrillar collagen scaffolds (BioBridge™, Fibralign Corp., Union City, CA, USA) to improve surgical strategies in patients with advanced lymphedema. Nanofibrillar collagen scaffolds mimic the collagen matrix that supports lymphatic vessels and can significantly promote lymphangiogenesis. Some preclinical and initial clinical studies suggest that the use of collagen scaffolds in routine surgery improves the outcome of microsurgical procedures in patients with lymphedema, showing promising results both in association with vascularized lymph node transplantation and LVA. Their capillary action aids the initiation of interstitial flow and appears to serve as a scaffold for endothelial cell attachment and alignment, leading to the restoration of small lymphatic ducts [67,82,83] (Figure 4).

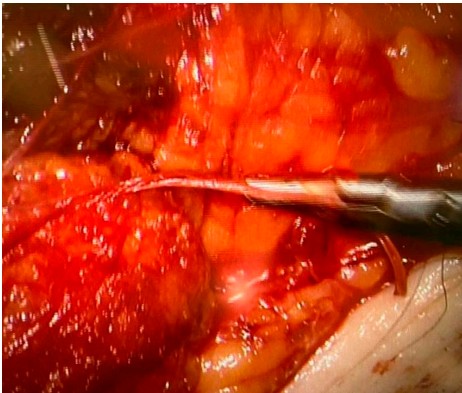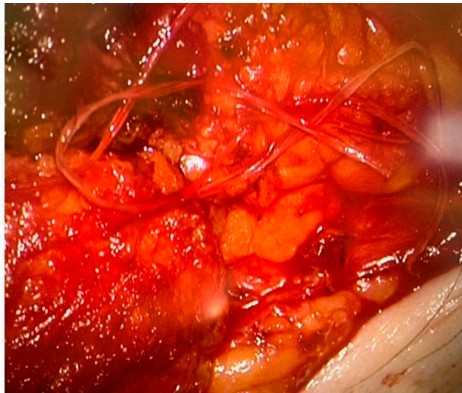

**Figure 4.** Two intraoperative images showing the use of Biobridge™ in a patient with secondary lymphedema.

*4.5. Robot Surgery*

According to a recent trial by van Mulken et al., super microsurgical LVA might be improved with the use of a microsurgical robot, but data are not sufficient now.

In general, although the mean duration of surgery appears to be significantly longer in robot-assisted surgeries compared to manual procedures, the time to completion of anastomosis decreased.

In conclusion, gathering data from recent studies, clinical outcomes comparing robot-assisted and manual LVA are similar, probably because robots used in present practice are not designed to be implied in super microsurgical procedures, although through tremor filtration and motion scaling, robot assistance could overcome limitations of the human hand in the near future [84–86].

**5. Future Perspectives**

Emerging diagnostic modalities may further improve the visualization of lymphatic vessels and offer insight into lymphedema pathogenesis and management approaches. For example, photoacoustic imaging is a relatively new ultrasound-based technique that has yet to be introduced into clinical practice. It utilizes the optical absorption properties of indocyanine green and offers high temporal and spatial resolution for the visualization of lymphatic vessels [49]. Compared to NIRF-L, photoacoustic imaging offers sharper images that are less affected by the thickness of subcutaneous tissues [87] and reliably identifies dermal backflow and functioning lymphatic vessels in areas with diffuse patterns on NIRF-L [88]. Although its use is currently limited by portability and imaging range, photoacoustic imaging can potentially improve our understanding of lymphatic anatomy and function in patients with lymphedema.

Improving the accessibility of lymphedema assessment through portable and easy-to-use equipment could greatly improve patients' quality of life and outcomes. Several options using three-dimensional optical imaging for the quantification of lymphedema have been proposed. In one study, a 3D camera was used for rapid assessment of secondary arm lymphedema in breast cancer patients. The results were comparable to that of manual circumference measurement and water displacement methods [88]. Another paper discussed the use of the infrared light depth sensor of XBOX Kinect with the Iterative Closest Points algorithm, which allowed robust volumetric assessment of lymphedema and could potentially be used at home with full autonomy after appropriate calibration [89]. In yet another step closer to the point-of-care lymphedema assessment, Yahathugoda et al. used an infrared sensor integrated with a computer table for a fully portable, fast, and reproducible assessment of lymphedema [90]. A recent study used a 3D infrared scanner for peri-operative follow-up of upper limb volume in breast cancer, with the potential for identifying patients at risk for lymphedema development [91].

The use of machine learning for segmentation [92], labeling, and quantification of imaging findings of lymphedema is an important direction that will define the future of lymphedema diagnosis and management. Automatization of these time and labor-consuming tasks will allow the optimization of the workflow of different imaging modalities, facilitate their routine use, and standardize the scoring systems results [49]. Several advanced image analysis solutions have already been proposed. An automated image segmentation protocol developed in IDL software was proposed for the assessment of lymphedema in breast cancer patients. The algorithm successfully segmented the epifascial and subfascial arm volumes on STIR and Dixon MRI acquisitions [93]. Nowak et al. described a deep-learning model for tissue segmentation based on the convolutional neural network EfficientNet-B1. The model successfully segmented subcutaneous and subfascial tissues on Dixon acquisitions (Dice score = $0.982 \pm 0.007$ and $0.989 \pm 0.003$, respectively) with a mean prediction time of 8 seconds. The segmented tissues were summarized in the visual format for the quantification of volume, distribution, and symmetry [94].

Artificial intelligence applications are common in the oncological field [95], and can also be used clinically for the prediction of lymphedema development. A fuzzy model algorithm was developed in 2011 by Vicentini et al. to objectively classify the risk and severity of lymphedema through a series of functional and clinical criteria. This approach facilitates standardization in lymphedema assessment and allows early rehabilitation and management of patients [96]. Fu et al. proposed an artificial neural network algorithm for the early detection of lymphedema in breast cancer patients based on real-time self-reported symptoms. The system achieved an accuracy of 93.75%, a sensitivity of 95.65%, and a specificity of 91.03%, with an AUC of 0.751 [97]. In another study, a machine-learning algorithm was used to identify the onset of edema through subtle changes in upper body motion range in breast cancer patients, as captured by a Kinect-based system [98]. Another machine-learning model allowed for accurately measuring arm volume in patients with lymphedema in a camera-like horizontal–vertical image scanning tool [99]. Kestenev et al. proposed a machine-learning model for classifying lymphedema based on the estimation of collagen disorganization in the skin with multiphoton laser microscopy. This non-invasive classification reached sensitivity and specificity of $0.79 \pm 0.11$ and $0.77 \pm 0.10$, respectively [100].

In the future, the increasing robustness of artificial intelligence algorithms and a growing understanding of lymphedema pathophysiology can facilitate the assessment of treatment progress and track the effectiveness of various therapies, providing clinicians with valuable insights into patient outcomes [97]. In the future, the increasing robustness of artificial intelligence algorithms and growing understanding of lymphedema pathophysiology can facilitate the assessment of treatment progress and track the effectiveness of various therapies, providing clinicians with valuable insights into patient outcomes.

## 6. Conclusions

In conclusion, lymphedema is a common complication of prostate cancer treatment that can have a significant impact on a patient's quality of life. Important progress has been achieved from a diagnostic point of view using magnetic resonance and from a therapeutic point of view with new conservative, medical, and surgical techniques.

Although there is currently no definitive cure for lymphedema, significant improvements in diagnosis and treatment are revolutionizing the approach to post-prostatectomy lymphedema patients. Artificial intelligence applications can play a significant role in the future for the prevention, diagnosis, and management of this disease.

**Author Contributions:** Conceptualization, M.C. (Michaela Cellina); Methodology, M.C. (Maurizio Cè); Literature research, O.C., L.M.G.B., G.I., E.D., G.D.P., and N.K.; Data Curation O.C., F.D.V., M.S., and A.M.; Writing—Original Draft Preparation, M.C. (Michaela Cellina), M.C. (Maurizio Cè), G.I., F.D.V., M.S., A.M., N.K., O.C., F.D.V., A.M., and M.S.; Writing—Review and Editing, M.C. (Michaela Cellina), M.C. (Maurizio Cè), L.M.G.B., G.I., E.D., G.D.P., M.C. (Maurizio Cè), F.D.V., O.C., and G.I.; Supervision, M.C. (Michaela Cellina), N.K., and M.C. (Maurizio Cè). All authors have read and agreed to the published version of the manuscript.

**Funding:** This research received no external funding.

**Institutional Review Board Statement:** Clinical images belong to the study approved by Comitato Etico Milano Area 1, protocol 47437/2019, approved on 7 November 2019.

**Conflicts of Interest:** The authors declare no conflict of interest.

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
