# Peer review of "Diagnosis and Treatment of Post-Prostatectomy Lymphedema: What’s New?"

_curroncol, doi:10.3390/curroncol30050341_

Round 1

Reviewer 1 Report

Dear Author

The topic is original, because Lymphedema is one of the important and serious side effects after prostatectomy. The authors could consider using more new articles.

This is a very good review study to investigate diagnosis and treatment of Lymphedema after radical prostatectomy surgery. 

Author Response

The authors thank the reviewer for the work and the positive comments.

Best regards

The authors

Reviewer 2 Report

The author comprehensively described about lower lymphedema from diagnosis to treatment.

I have some suggestion and references to add this review manuscript.

1. Abstract

The author noted lymphovenous bypass (LVB) and LVA.

Please unify the writing.

2. Abstract

The author mentioned about BioBridge in conclusion at abstract and artificial intelligence in conclusion at main text.

What is the most important and advocating conclusion in this manuscript.

Not only BioBridge and AI have an important role to treat lymphedema.

And the title "post-prostatectomy" is not correlated with conclusion.

2. Line 62

The title of this manuscript is including "post-prostatectomy".

Is the etiology of primary lymphedema necessary?

Please focus on "post-prostatectomy".

e.g. are there any difference of incidence rate between post-prostatectomy and post-hysterectomy?

3. Line 170

What is NCMRL?

Non contrast MR lymphography? Please spell out.

4.  Line 257-312

The author described CDT and medication in Consevative treatments.

Please dived the manual therapy and medication as well as LVA and VLNT.

I suggest these manuscripts are helpful for your manuscript.

S. G. Rockson et al. Pilot studies demonstrate the potential benefits of antiinflammatory therapy in human lymphedema. JCI Insight 2018 Vol. 3 Issue 20 

This manuscript reported the efficacy of LTB4 inhibitors.

B. J. Mehrara et al. Pilot Study of Anti-Th2 Immunotherapy for the Treatment of Breast Cancer-Related Upper Extremity Lymphedema. Biology (Basel) 2021 Vol. 10 Issue 9 

This manuscript reported the efficacy of Th2 inhibition with neutralizing antibodies.

5. Line 343

The author mentioned about liposuction and subsequently continuing conservative therapy must needed.

Please indicate the evidence and reference.

6. Line 345

Combined surgery with LVA, VLNT and liposuction is effective in staged lymphedema for downsizing of affecting limb.

Please add the reference.

H. Imai, et al. Correlation between Lymphatic Surgery Outcome and Lymphatic Image-Staging or Clinical Severity in Patients with Lymphedema. Journal of Clinical Medicine 2022 Vol. 11 Issue 17 Pages 4979

7. Line 458- 507

Combined VLNT and VEGF-C treatment was reported. This manuscript will help your manuscript to make value.

I. M. Leppapuska et al. Phase 1 Lymfactin(R) Study: 24-month Efficacy and Safety Results of Combined Adenoviral VEGF-C and Lymph Node Transfer Treatment for Upper Extremity Lymphedema. J Plast Reconstr Aesthet Surg 2022 Vol. 75 Issue 11 Pages 3938-3945.

Author Response

The authors thank the reviewer for the work and precious suggestions.

We performed all the required changes as follows:

1. Abstract

The author noted lymphovenous bypass (LVB) and LVA.

Please unify the writing.

Thank you for the suggestion. We homogenize the terminology

2. Abstract

The author mentioned about BioBridge in conclusion at abstract and artificial intelligence in conclusion at main text.

What is the most important and advocating conclusion in this manuscript.

Not only BioBridge and AI have an important role to treat lymphedema.

And the title "post-prostatectomy" is not correlated with conclusion.

Thank you for the suggestion, we modified the manuscript accordingly

2. Line 62

The title of this manuscript is including "post-prostatectomy".

Is the etiology of primary lymphedema necessary?

Please focus on "post-prostatectomy".

e.g. are there any difference of incidence rate between post-prostatectomy and post-hysterectomy?

We modified the text and focused on the post-prostatectomy lymphedema

3. Line 170

What is NCMRL?

Non contrast MR lymphography? Please spell out.

We apologize and explained the acronym

4.  Line 257-312

The author described CDT and medication in Consevative treatments.

Please dived the manual therapy and medication as well as LVA and VLNT.

I suggest these manuscripts are helpful for your manuscript.

S. G. Rockson et al. Pilot studies demonstrate the potential benefits of antiinflammatory therapy in human lymphedema. JCI Insight 2018 Vol. 3 Issue 20 

This manuscript reported the efficacy of LTB4 inhibitors.

B. J. Mehrara et al. Pilot Study of Anti-Th2 Immunotherapy for the Treatment of Breast Cancer-Related Upper Extremity Lymphedema. Biology (Basel) 2021 Vol. 10 Issue 9 

This manuscript reported the efficacy of Th2 inhibition with neutralizing antibodies.

Thank you for the suggestion. We inserted the above cited articles in our references

5. Line 343

The author mentioned about liposuction and subsequently continuing conservative therapy must needed.

Please indicate the evidence and reference.

Thank you for the suggestion, we added an appropriate reference

6. Line 345

Combined surgery with LVA, VLNT and liposuction is effective in staged lymphedema for downsizing of affecting limb.

Please add the reference.

H. Imai, et al. Correlation between Lymphatic Surgery Outcome and Lymphatic Image-Staging or Clinical Severity in Patients with Lymphedema. Journal of Clinical Medicine 2022 Vol. 11 Issue 17 Pages 4979

Thank you for the suggestion, we added the reference

7. Line 458- 507

Combined VLNT and VEGF-C treatment was reported. This manuscript will help your manuscript to make value.

I. M. Leppapuska et al. Phase 1 Lymfactin(R) Study: 24-month Efficacy and Safety Results of Combined Adenoviral VEGF-C and Lymph Node Transfer Treatment for Upper Extremity Lymphedema. J Plast Reconstr Aesthet Surg 2022 Vol. 75 Issue 11 Pages 3938-3945.

Thank you for the suggestion. We inserted all the above references

Thank you

Best regards

The authors

Reviewer 3 Report

The authors present a very well written review article that brings the reader up to date on the various modalities for the diagnosis and treatment of lymphedema. The article is well organized and covers most of the developments in the field of lymphedema treatment. The suggestions I have include minor changes and some additional references and information that can benefit the article even more:

1.     Line 55 – TNF-a

2.     Line 183 – “…a 1.5-T” please explain this in a little more detail, since not all readers may be familiar with MRI terminologies and abbreviations.

3.     Line 189 – “T2 weighted sequence”, please provide some extra explanation here as well.

4.     Fig. 2: Please use a better image for ICG imaging. There are better images available from multiple papers.

5.     Better alternatives to ICG have been proposed for NIR imaging by Weiler et al. (https://doi.org/10.3389/fphys.2013.00215) and the authors might consider including it in the paper.

6.     When discussing conservative treatments, the authors can also consider citing the paper by Mukherjee et al. (https://doi.org/10.1113/JP281206) that looks at the impact of parameters of intermittent pneumatic compression on lymphatic contractility, showing that optimal stimulation parameters are possible to maximize lymphatic transport function.

7.      There are multiple articles that look at measurement of lymphedema arm volume using commercial infrared based 3d scanning devices. These can be discussed in the future perspectives section. Some examples are https://doi.org/10.1113/JP281206 , https://doi.org/10.4269%2Fajtmh.17-0504 , https://doi.org/10.1093/ptj/pzz175 .

Author Response

The authors thank the reviewer for the work and precious suggestions.

We modified the manuscript accordingly

The authors present a very well written review article that brings the reader up to date on the various modalities for the diagnosis and treatment of lymphedema. The article is well organized and covers most of the developments in the field of lymphedema treatment. The suggestions I have include minor changes and some additional references and information that can benefit the article even more:

1.     Line 55 – TNF-a

2.     Line 183 – “…a 1.5-T” please explain this in a little more detail, since not all readers may be familiar with MRI terminologies and abbreviations.

Thank you. We added the required explainations

3.     Line 189 – “T2 weighted sequence”, please provide some extra explanation here as well.

Thank you. We added the required explainations

4.     Fig. 2: Please use a better image for ICG imaging. There are better images available from multiple papers.

We replaced the image, as suggested

5.     Better alternatives to ICG have been proposed for NIR imaging by Weiler et al. (https://doi.org/10.3389/fphys.2013.00215) and the authors might consider including it in the paper.

We included the suggested reference in the paper

6.     When discussing conservative treatments, the authors can also consider citing the paper by Mukherjee et al. (https://doi.org/10.1113/JP281206) that looks at the impact of parameters of intermittent pneumatic compression on lymphatic contractility, showing that optimal stimulation parameters are possible to maximize lymphatic transport function.

We included the suggested reference in the paper

7.      There are multiple articles that look at measurement of lymphedema arm volume using commercial infrared based 3d scanning devices. These can be discussed in the future perspectives section. Some examples are https://doi.org/10.1113/JP281206 , https://doi.org/10.4269%2Fajtmh.17-0504 , https://doi.org/10.1093/ptj/pzz175 .

We included the suggested reference in the paper

Thank you very much

Best regards

The authors

Round 2

Reviewer 2 Report

The manuscript is much improved. However, I couldn't find the author point-by-point response.

Author Response

I am sorry for the problem.

The response was the following:

The authors thank the reviewer for the work and precious suggestions.

We performed all the required changes as follows:

1. Abstract

The author noted lymphovenous bypass (LVB) and LVA.

Please unify the writing.

Thank you for the suggestion. We homogenize the terminology

2. Abstract

The author mentioned about BioBridge in conclusion at abstract and artificial intelligence in conclusion at main text.

What is the most important and advocating conclusion in this manuscript.

Not only BioBridge and AI have an important role to treat lymphedema.

And the title "post-prostatectomy" is not correlated with conclusion.

Thank you for the suggestion, we modified the manuscript accordingly

2. Line 62

The title of this manuscript is including "post-prostatectomy".

Is the etiology of primary lymphedema necessary?

Please focus on "post-prostatectomy".

e.g. are there any difference of incidence rate between post-prostatectomy and post-hysterectomy?

We modified the text and focused on the post-prostatectomy lymphedema

3. Line 170

What is NCMRL?

Non contrast MR lymphography? Please spell out.

We apologize and explained the acronym

4.  Line 257-312

The author described CDT and medication in Consevative treatments.

Please dived the manual therapy and medication as well as LVA and VLNT.

I suggest these manuscripts are helpful for your manuscript.

S. G. Rockson et al. Pilot studies demonstrate the potential benefits of antiinflammatory therapy in human lymphedema. JCI Insight 2018 Vol. 3 Issue 20 

This manuscript reported the efficacy of LTB4 inhibitors.

B. J. Mehrara et al. Pilot Study of Anti-Th2 Immunotherapy for the Treatment of Breast Cancer-Related Upper Extremity Lymphedema. Biology (Basel) 2021 Vol. 10 Issue 9 

This manuscript reported the efficacy of Th2 inhibition with neutralizing antibodies.

Thank you for the suggestion. We inserted the above cited articles in our references

5. Line 343

The author mentioned about liposuction and subsequently continuing conservative therapy must needed.

Please indicate the evidence and reference.

Thank you for the suggestion, we added an appropriate reference

6. Line 345

Combined surgery with LVA, VLNT and liposuction is effective in staged lymphedema for downsizing of affecting limb.

Please add the reference.

H. Imai, et al. Correlation between Lymphatic Surgery Outcome and Lymphatic Image-Staging or Clinical Severity in Patients with Lymphedema. Journal of Clinical Medicine 2022 Vol. 11 Issue 17 Pages 4979

Thank you for the suggestion, we added the reference

7. Line 458- 507

Combined VLNT and VEGF-C treatment was reported. This manuscript will help your manuscript to make value.

I. M. Leppapuska et al. Phase 1 Lymfactin(R) Study: 24-month Efficacy and Safety Results of Combined Adenoviral VEGF-C and Lymph Node Transfer Treatment for Upper Extremity Lymphedema. J Plast Reconstr Aesthet Surg 2022 Vol. 75 Issue 11 Pages 3938-3945.

Thank you for the suggestion. We inserted all the above references

Thank you

Best regards

The authors